# FEDERATED LEARNING NODES CAN RECONSTRUCT PEERS' IMAGE DATA

## ABSTRACT

Federated learning (FL) is a privacy-preserving machine learning framework that enables multiple nodes to train models on their local data and periodically average weight updates to benefit from other nodes' training. Each node's goal is to collaborate with other nodes to improve the model's performance while keeping its training data private. However, this framework does not guarantee data privacy. Prior work has shown that the gradient-sharing steps in FL can be vulnerable to data reconstruction attacks from a honest-but-curious central server. In this work, we show that a honest-but-curious node/client can also launch attacks to reconstruct peers' image data in a centralized system, presenting a severe privacy risk. We demonstrate that a single client can silently reconstruct other clients' private images using diluted information available within consecutive updates. We leverage state-of-the-art diffusion models to enhance the perceptual quality and recognizability of the reconstructed images, further demonstrating the risk of information leakage at a semantic level. This highlights the need for more robust privacy-preserving mechanisms that protect against silent client-side attacks during federated training.

## 1 INTRODUCTION

Federated learning (FL) has attracted significant attention as a promising approach to privacy-preserving machine learning (McMahan et al., 2017; Kairouz et al., 2021). In this framework, a central server coordinates training by multiple clients. In each training round, the server broadcasts a shared model to a subset of clients. Each client computes a gradient by training the model on its private data and returns the gradient to the server. The server then averages all the gradients and updates the model. This approach enables each participant to benefit from a model trained on more data without sharing its own data. FL has the potential to revolutionize collaborative efforts in such real-world applications as healthcare and finance, enabling participants to train better models without compromising data privacy (Li et al., 2020a).

Despite the intent to protect privacy through FL, prior works have shown that a honest-but-curious central server can reconstruct a client's training data. This is done by adjusting a dummy input to the model until its resulting gradient closely matches the gradient sent by the client (Zhu et al., 2019; Geiping et al., 2020; Yue et al., 2023). Meanwhile, studies on malicious clients have shown that a client can disrupt federated training by sending adversarial data to the server (Blanchard et al., 2017; Shi et al., 2022). However, far less attention has been given to the potential for clients to reconstruct others' data while honestly participating in the FL network.

Our work explores the extent to which a single client can reconstruct peers' training data while adhering to FL protocols. We introduce a novel client-to-client attack depicted in Figure 1, where the attacker exploits weight updates between consecutive training rounds to reconstruct training images. By participating in two consecutive training rounds and comparing the global model's weights, a single client extracts the averaged gradient of all participants in the earlier round. Unlike server-side gradient inversion (Zhu et al., 2019; Geiping et al., 2020; Yue et al., 2023), this attack requires isolating individual data from a diluted mixture of gradient updates. Despite this challenge, we show that the attacker is able to reconstruct images from every other client.

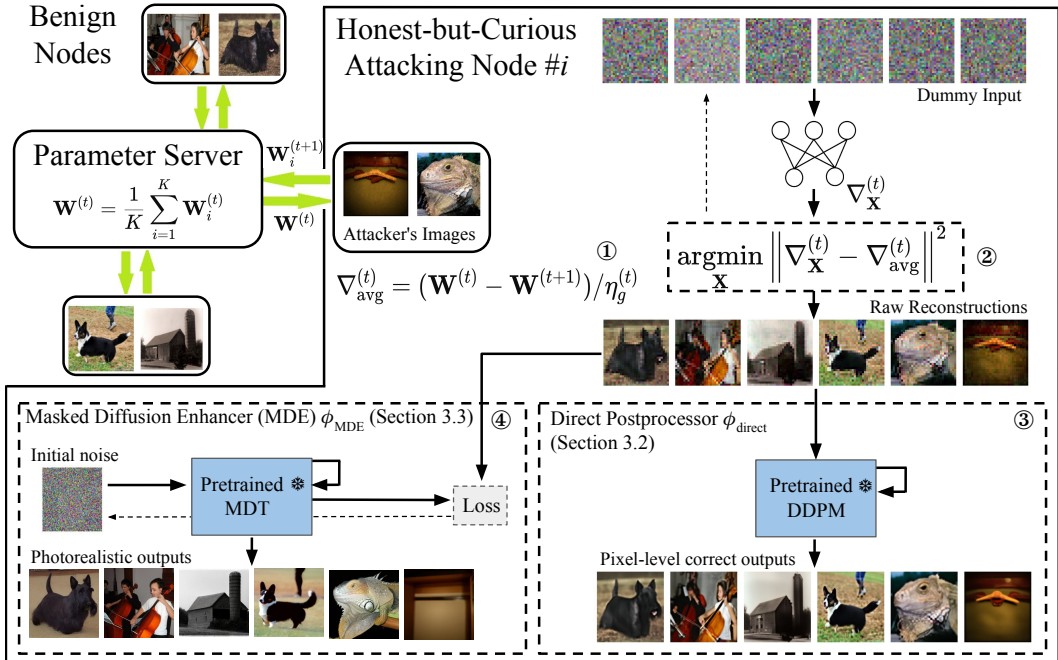

Figure 1: Overview of the proposed honest-but-curious client attack. The attacker participates in two consecutive training rounds to obtain the global model's gradient update by ① differencing the model weights. The attacker then ② inverts the update to obtain each client's training data. To address the challenge of recovering high-quality images from diluted information hidden in the global gradient update, the raw reconstructions are postprocessed using either ③ a direct technique respecting pixel-level correctness or ④ a semantic technique focusing on producing photorealistic images.

We further propose to utilize two image postprocessing techniques based on diffusion models to improve the quality and recognizability of attack's raw reconstructed images. Our first technique uses a pretrained masked diffusion transformer (MDT) (Gao et al., 2023) to generate high quality images that resemble the raw attack results on a semantic level. Our second technique uses denoising diffusion probabilistic models (DDPMs) (Ho et al., 2020) to enhance the raw reconstructions at the pixel level through super resolution and denoising (Kawar et al., 2022). This paper's contributions are threefold.

1. We demonstrate that clients participating in FedAvg (McMahan et al., 2017) can exploit the model updating process to reconstruct peers' data, which reveals a previously unrecognized privacy risk in FL.

2. The proposed masked diffusion enhancer (MDE) generates sharp, high-resolution images from the low-resolution, color-aliased raw reconstructions. The generated images resemble a target image on a semantic level, preserving its geometric shape and perceptual features with photorealistic quality.

3. The proposed DDPM-based image postprocessing simultaneously denoises and upsamples raw reconstructed images. This improves image resolution and object recognizability, achieving strong pixel-wise similarity to ground-truth images.

## 2    RELATED WORK

**Server-Side Gradient Inversion.** The assumption that FL inherently protects data privacy has been challenged by studies exposing vulnerabilities to gradient inversion attacks from the central server (Zhu et al., 2019; Geiping et al., 2020). These attacks exploit the gradients shared by clients to reconstruct private training data. They revealed that by iteratively updating a dummy input to produce a gradient similar to a given target gradient, the server could generate images closely resembling the participant's original training data. Various defense mechanisms have been proposed to protect

against these attacks, including gradient compression, perturbation, and differential privacy techniques (Zhang et al., 2020; Sun et al., 2021). Despite these efforts, recent studies have shown that these defenses may not effectively prevent training data from being meaningfully reconstructed. For example, Yue et al. (2023) overcame these defense methods by leveraging latent space reconstruction and incorporating generative models to remove distortion from reconstructed images.

**Client-Side Model Inversion.** While the majority of research has focused on server-side attacks, the potential for client-side attacks has been less examined. Wu et al. (2024) investigated model inversion attacks, where a client exploits the model's overfitting to reconstruct training data. This approach relies on manipulating the model rather than directly reconstructing other clients' data. Similar to our work's takeaway, their results demonstrated that clients can reconstruct peers' images without disrupting the training process. However, due to the challenging nature of model inversion, their method produces reconstructed images far less similar to the target than those from gradient inversion attacks.

**Malicious Client Attacks.** A parallel research direction focuses on attacks where a malicious client interferes with the FL process (Blanchard et al., 2017; Shejwalkar et al., 2022). Specifically, malicious clients can manipulate the model updates by using poisoned data or sending poisoned gradients to the server to impede convergence. Meanwhile, researchers have shown that malicious modifications can compromise privacy easily (Fowl et al., 2021; Wen et al., 2022). While this introduces unique security challenges in FL, our attack does not disrupt the training process and is difficult to be detected by the server or other clients.

## 3 PROPOSED IMAGE RECONSTRUCTION ATTACK BY CURIOUS CLIENTS

In this section, we present the gradient inversion attack, which allows an honest-but-curious client to reconstruct image data from other clients. To enhance this reconstruction, two postprocessing methods are introduced to achieve both fine-grained quality and perceptual realism. The first postprocessing method improves the images at the pixel level with enhanced details. The second method, built on a masked diffusion enhancer, refines the images at the semantic level and produces photo-realistic reconstructions.

### 3.1 ATTACK FRAMEWORK

**Threat Model.** We consider an honest-but-curious client (or curious client, for simplicity) attacker. It aims to reconstruct other clients' training data while following the protocol of FL. The attacker does not disrupt the model training process. The curious client does not have direct access to the gradients from other clients. However, it receives an updated version of the shared model from the server at each communication round. Additionally, we assume as in Li et al. (2020b); Huang et al. (2020) that all client updates in a given round have been computed using the same learning rate, which is applied locally if each client transmits a model update, as shown in Eq. (2). It may also be applied globally if clients transmit raw gradients to the server, as discussed in Section 4. The attacker may not know the number of clients in each training round but can correctly guess the total number of training images. We follow the assumption of Yue et al. (2023) that each client trains for $\tau$ iterations on the same minibatch of images in each local iteration/update round and that the class labels have been analytically inverted as in Ma et al. (2023). We target cross-silo FL scenarios, in which a small number of clients collaborate to overcome data scarcity. For example, a group of hospitals may use FL to develop a classifier for rare diseases from CT scans, where each has limited training examples and images cannot be directly shared due to privacy concerns. We assume that the system is designed to prioritize model accuracy and uses synchronous gradient updates. Clients are not edge devices and have sufficient computational resources to perform the optimization process while participating in FL.

We describe the FL process to be attacked as follows. The $k$th client at time $t$ uses the same minibatch of size $N_k$ to compute its local weights $\mathbf{W}_k^{(t,u)}$ across all local iterations $u$ until $u = \tau$, where $\tau$ is the number of local training iterations. Each client's final local weight can be expressed as:

$$\mathbf{W}_k^{(t,\tau)} = \mathbf{W}^{(t)} - \frac{\eta_\mathbf{g}}{N_k}\Delta_k^{(t)}, \qquad \Delta_k^{(t)} = \sum_{u=0}^{\tau-1}\sum_{i=1}^{N_k} \nabla\ell(\mathbf{W}_k^{(t,u)}; \mathbf{X}_{k,i}; \mathbf{Y}_{k,i}), \qquad (1)$$

where $\mathbf{W}^{(t)}$ is the global model parameters at time $t$, $\nabla \ell(\cdot)$ is the gradient of the loss function, the doubly indexed $\mathbf{X}_{k,i} \in \mathbb{R}^{C \times H \times W}$, $\mathbf{Y}_{k,i} \in \mathbb{R}$ are the $i$th training image and label from client $k$, respectively, and $C$, $H$, and $W$ are the number of channels, the height, and the width of the images. The server generates the global weights by a weighted average of all clients' final local weights, namely, $\mathbf{W}^{(t+1)} = \frac{1}{N} \sum_{k=1}^{K} N_k \mathbf{W}_k^{(t,\tau)}$, where $N = \sum_{k=1}^{K} N_k$ is the total number of training examples across $K$ clients, with each client having a fixed minibatch of $N_k$ images. Substituting $\mathbf{W}_k^{(t,\tau)}$ into the expression for $\mathbf{W}^{(t+1)}$, we obtain the global weight update equation:

$$\mathbf{W}^{(t+1)} = \mathbf{W}^{(t)} - \frac{\eta_g}{N} \Delta_k^{(t)}. \tag{2}$$

We note that scaling each client's update by its number of training images $N_k$ causes the gradient of each training image $\mathbf{X}_{k,i}$ to be weighted equally in the global update.

Our approach to reconstructing data from the global model updates builds upon traditional gradient inversion and includes extra initialization and calculation steps to separate individual training images from the averaged global update. Our attacker engages in two consecutive rounds of FedAvg and obtains two versions of the global model parameters, $\mathbf{W}^{(t)}$ and $\mathbf{W}^{(t+1)}$. By computing the difference between successive model weights, the attacker can infer the gradient used for the global model update: $\nabla_{\text{avg}}^{(t)} = (\mathbf{W}^{(t)} - \mathbf{W}^{(t+1)})/\eta_g^{(t)}$, where $\eta_g^{(t)}$ is the globally-determined learning rate for round $t$.

To reconstruct training images, our attacker initializes dummy image data $\mathbf{X} \in \mathbb{R}^{N \times C \times H \times W}$ and labels $\mathbf{Y} \in \mathbb{R}^N$. The attacker passes them through a global model and compares the resulting gradient update $\Delta^{(t)}(\mathbf{X}, \mathbf{Y}) = \sum_{u=0}^{\tau-1} \sum_{l=1}^{N} \nabla \ell(\mathbf{W}^{(t,u)}; \mathbf{X}_l; \mathbf{Y}_l)$ to the target gradient $\nabla_{\text{avg}}^{(t)}$, where the singly indexed $\mathbf{X}_l$ and $\mathbf{Y}_l$ are the $l$th dummy image and label for the combined dataset. Following the gradient inversion framework, the goal is to iteratively refine $\mathbf{X}$ until it closely approximates the data used to compute the target gradient. The attacker solves the following optimization problem:

$$\hat{\mathbf{X}} = \arg \min_{\mathbf{X}} \left\| \Delta^{(t)}(\mathbf{X}, \mathbf{Y}) - \nabla_{\text{avg}}^{(t)} \right\|^2, \tag{3}$$

where the evolving global model $\{\mathbf{W}^{(t,u)}\}_{u=0}^{\tau-1}$ requires only the knowledge of the total number of images, eliminating the need to know the number of clients and the image counts from all clients.

Finally, the attacker applies a postprocessing function $\phi(\cdot)$ to improve the quality of the reconstructed images $\tilde{\mathbf{X}} = \phi(\hat{\mathbf{X}})$. This shows that a curious client attacker is able to follow an approach similar to server-side gradient inversion and obtain reconstructed data from all other clients from only two consecutive versions of the model weights. We describe below two methods of postprocessing $\hat{\mathbf{X}}$ to improve its quality at either a pixel or semantic level.

## 3.2 Direct Postprocessing for Pixel-Level Image Enhancement

To reconstruct the target data more effectively, we introduce a direct postprocessing method that utilizes pretrained diffusion models to perform super resolution and denoising on the raw image reconstructions. The raw reconstructed images from the attack may be low-resolution or have pixel artifacts due to imperfect gradient inversion. This problem may also be more severe in our attack compared to server-side gradient inversion as the target gradient contains diluted information from multiple clients. To address this problem, we introduce a postprocessing implementation, $\phi(\cdot) \equiv \phi_{\text{direct}}(\cdot)$ that uses pretrained diffusion models to directly postprocess the raw reconstructed images. Diffusion models have demonstrated good performance in image generation and restoration tasks and are able to produce more realistic images with a lower likelihood of hallucination (Dhariwal & Nichol, 2021). Our method follows the denoising diffusion restoration models (DDRM) framework and utilizes a pretrained DDPM (Ho et al., 2020) as a backbone model. DDRM has demonstrated strong performance across various image restoration tasks, including super resolution and denoising (Kawar et al., 2022). By increasing resolution and removing noise, we aim to accurately reveal details of the ground truth images and make the reconstructions more recognizable.

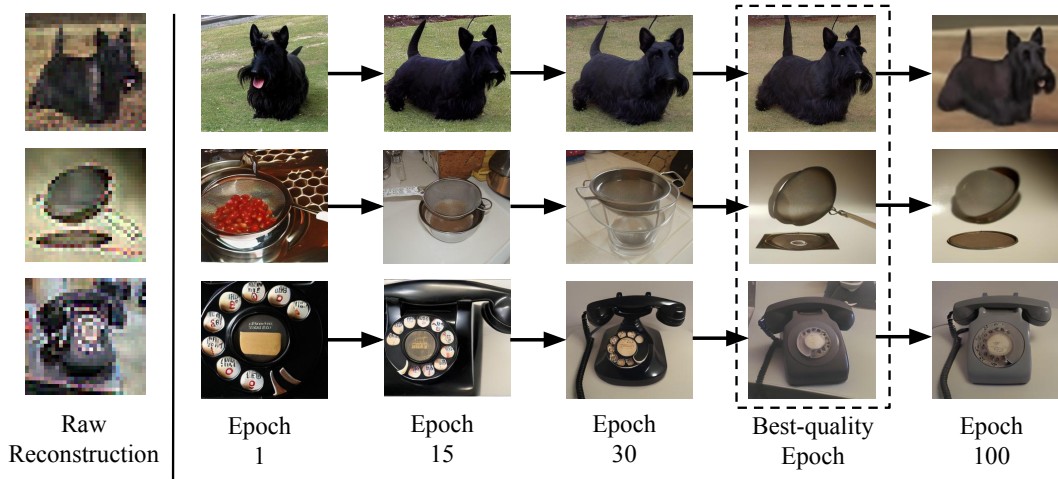

Raw Reconstruction | Epoch 1 | Epoch 15 | Epoch 30 | Best-quality Epoch | Epoch 100

Figure 2: Photorealistic images reconstructed by the proposed semantic reconstruction method MDE. This diffusion-based method iteratively refines its generated image by referring to the raw reconstruction. As iterations progress, the image increasingly assumes the shape and perceptual features of the raw reconstruction. After some optimal epoch number (determined by visual inspection of the attacker), the reconstructed image strongly resembles the target at a high quality. Beyond this point, further optimization may produce blurry images due to overfitting.

### 3.3 MASKED DIFFUSION ENHANCER: RECONSTRUCTION AT A SEMANTIC LEVEL

In this subsection, we also introduce a method to reconstruct target images at the semantic level, $\phi(\cdot) \equiv \phi_{\text{semantic}}(\cdot)$, the masked diffusion enhancer (MDE). The goal of MDE is to generate sharp, high-resolution images from the low-resolution, color-aliased raw attack results. This approach complements the direct postprocessing technique, as the generated images resemble the raw reconstructions at the semantic level, rather than at the pixel level. The generated images preserve the shape and perceptual features of the target image while achieving photorealistic quality.

**Backbone Model.** We use masked diffusion transformer (MDT) as the backbone of our reconstruction technique. MDT has been proven to achieve state-of-the-art performance in image generation (Gao et al., 2023). Due to its extensive training and flexibility, MDT has learned a complex representation of each image class that enables it to accurately reconstruct each image's semantic features through projection onto the manifold. Following the diffusion framework, MDT generates images by starting from a Gaussian noise vector $\mathbf{X}_T \sim \mathcal{N}(\mathbf{0}, \mathbf{I})$, where $T$ is the total number of diffusion steps. At each step $t$, the model predicts a noise residual $\epsilon_\theta(\mathbf{X}_t)$, and uses it to refine the noisy image $\mathbf{X}_t$ to $\mathbf{X}_{t-1}$. After $T$ iterations, the initial noise vector $\mathbf{X}_T$ will be transformed into a high quality image $\mathbf{X}_0$. For our reconstruction technique, we leverage a pretrained MDT and freeze its model parameters throughout the process to maintain consistency in the image generation pipeline.

**Proposed Masked Diffusion Enhancer (MDE).** MDE optimizes the initial noise vector $\mathbf{X}_T$ to generate an image that closely matches a target image $\hat{\mathbf{X}}$. During optimization, the noise predictions $\epsilon_\theta(\mathbf{X}_t)$ at each timestep are treated as constants. The objective of MDE is to minimize the mean squared error (MSE) between the final generated image $\mu_\theta(X_T, T)$ and the target image $\hat{\mathbf{X}}$:

$$\tilde{\mathbf{X}}_T = \arg\min_{\mathbf{X}_T} \left\| \mu_\theta(\mathbf{X}_T, T) - \hat{\mathbf{X}} \right\|_2^2, \tag{4}$$

where $\mu_\theta(\mathbf{X}_T, T)$ denotes the final image produced from the initial noise vector $\mathbf{X}_T$ after all diffusion steps. By optimizing $\mathbf{X}_T$ based on the loss term, we guide the model to generate images that have the same shape and perceptual features as the target image.

## 4 EXPERIMENTAL RESULTS

This section first presents the performance of the proposed reconstruction attack in terms of image reconstruction quality against gradients averaged from multiple clients. Factors affecting recon-

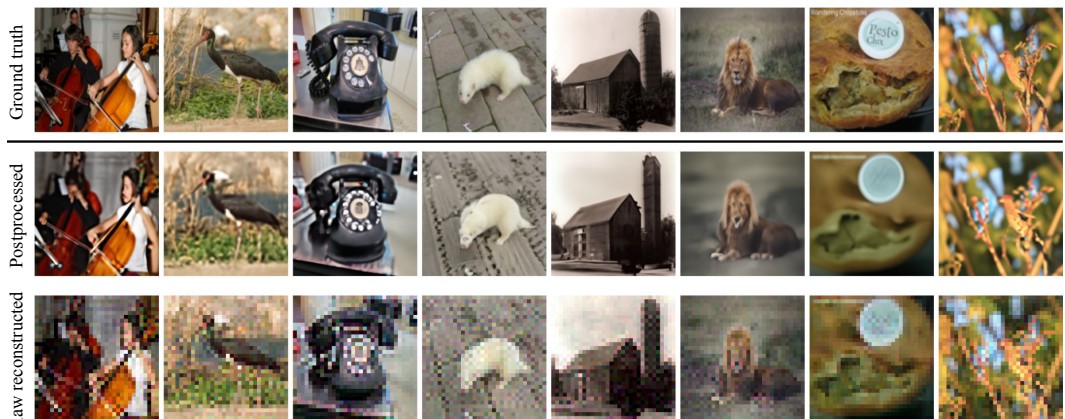

Figure 3: Pixel-level correct images reconstructed by the proposed attack before and after direct postprocessing (Section 3.2). The second and third rows show postprocessed and raw reconstructed images. The raw reconstruction results from our attack are low-resolution and have significant color aliasing. Our direct postprocessing method increases the resolution and simultaneously denoises the images while maintaining pixel-level correctness, revealing image details that make the reconstructions easier to recognize.

struction quality, including the number of local iterations and client batch size, will be analyzed. The postprocessing modules will be ablated to examine their benefits on image reconstruction. The state of the art will be compared and the limitation of the proposed attack will be discussed.

**Experimental Conditions.** We evaluate our reconstruction attacks using the ImageNet (Deng et al., 2009) and MNIST (LeCun et al., 1998) datasets. We employ LeNet (LeCun et al., 1998) and ResNet (He et al., 2016) as the global models and conduct experiments under the FedAvg framework. Each client performs 3 iterations of local training on 16 images as this batch size provides a baseline where the attack reconstructs recognizable images from the target gradient. As more clients participate in training, the training image count from the global model's perspective increases proportionally. The attacker uses a learning rate of 0.1 to optimize the dummy data and the attack is conducted after the first FL round, following the approach of Yue et al. (2023). Before inverting the target gradient, the attacker encodes its dummy data through bicubic sampling with a scale factor of 4 to reduce the number of unknown parameters. This has been proven to save convergence time and improve image quality in gradient inversion (Yue et al., 2023). We use LPIPS (Zhang et al., 2018) as the primary metric to evaluate quality of the attack's reconstructed images as it provides the best representation of perceptual image quality based on our experiments, through we observe similar trends for SSIM (Wang et al., 2004) and PSNR/MSE.

**Main Results.** Figure 2 demonstrates that MDE effectively transforms low-resolution, color-aliased raw reconstructions into sharp, high-resolution images. The model was provided with randomly selected reconstructions from an attack on a system with four clients and iteratively refined the outputs over 100 epochs. In the first epoch, the output is a random image from the target class. As optimization progresses, the generated images increasingly resemble the target. We observe that at an optimal epoch number, the output images closely match the target, preserving its geometric structure and perceptual features with photorealistic quality. This optimal point varies across target images and was determined qualitatively based on the raw reconstructed images. Beyond this point, although the generated images continue to match the target semantically, their quality degrades,

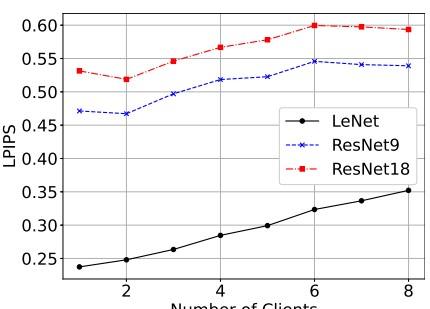

Figure 4: LPIPS of reconstructed images vs. number of clients with three different models: LeNet5, ResNet9, and ResNet18.

becoming blurrier. We attribute this to the model overfitting that learns the pixelation and blurriness of the target to minimize the MSE loss.

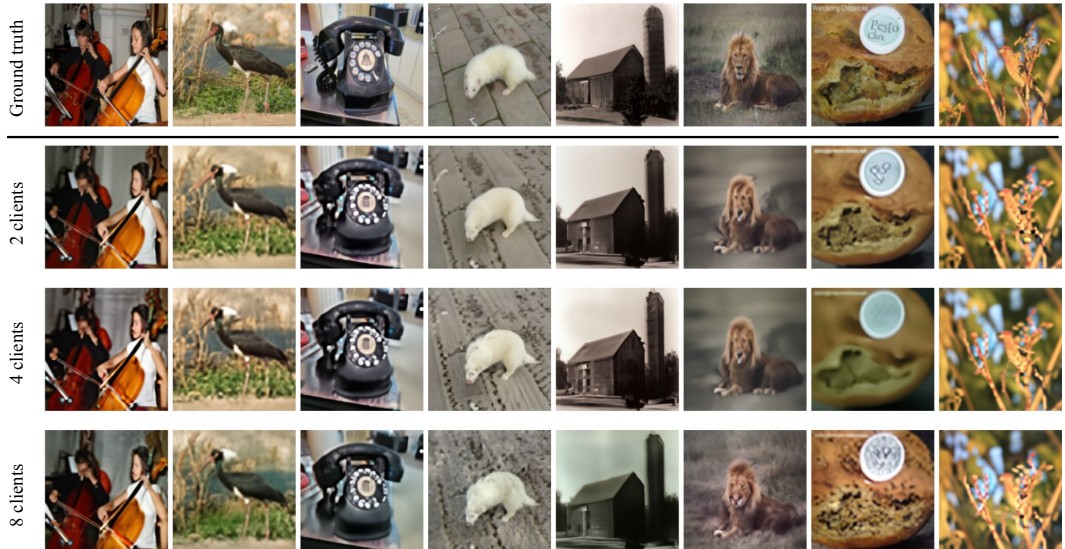

Figure 5: Pixel-level correct images reconstructed from the proposed honest-but-curious client-based attack. Rows 2–4 show reconstructions from gradients averaged across 2, 4, and 8 clients using LeNet5 as the global model with 16 images per client and 3 local training iterations. The images remain high quality even when the attack is performed against multiple clients.

Figure 3 illustrates the impact of the proposed DDPM-based direct image postprocessing in enhancing the quality of the raw reconstructions while preserving pixel-wise accuracy. The raw reconstructed images, constrained by the dummy data's encoding, are $32{\times}32$ pixels compared to the $128{\times}128$ ground-truth images and exhibit pixel artifacts and color aliasing due to imperfect reconstruction. Our direct postprocessing method simultaneously performs super-resolution and denoising, addressing these quality issues. The resulting images are sharper, more recognizable, and retain details that closely match the ground truth, significantly improving resolution and object recognizability over the raw reconstructions. Figure 5 shows reconstructed images from attacks against systems with 2, 4, and 8 clients. As the number of clients increases, the reconstruction task becomes more difficult but we observe that our attack is still able to effectively reconstruct images from the target gradient. However, we observe a gradual decline in image reconstruction quality, measured by LPIPS (Zhang et al., 2018) and SSIM (Wang et al., 2004) of the reconstructed images, as shown in Figure 4. With a larger number of clients, the initial reconstructions exhibit increasing levels of noise and color aliasing. This trend is consistent across a range of global models because the information contained within the target gradient becomes increasingly diluted as it is averaged from more clients.

**Ablation Study.** We examine how much impact the direct and semantic postprocessing blocks have on the quality of the attacker's reconstructed images. Figure 6 shows that directly postprocessing the raw reconstructed images results in a 20–30% improvement in LPIPS for systems with 2–8 clients. Figure 3 visually compares reconstructed images from a system with four clients before and after direct postprocessing. The raw images are low resolution and may be somewhat difficult to recognize while the postprocessed versions show much finer details and have recognizable features. This demonstrates the utility of our direct postprocessing technique in increasing the pixel-wise accuracy and recognizability of the reconstructed images.

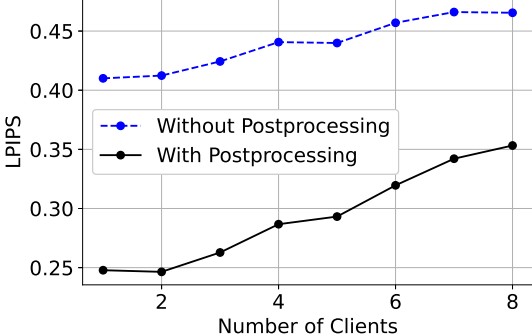

Figure 6: LPIPS of reconstructed images with varying number of clients. Our direct postprocessing technique significantly improves reconstruction quality compared to the raw attack results.

Additionally, Figure 2 shows the effect of postprocessing the raw reconstructed images using the proposed MDE. The final results have the same shape and perceptual features of the raw reconstructions without any pixel artifacts, color aliasing, or blurriness. However, MDE's goal is not to achieve pixel-wise accuracy so the generated images should not be compared quantitatively to the raw reconstructions.

**Factor Analysis.** Figure 7 reveals the effect of local iterations and client batch size on image reconstruction quality. These factors directly influence the attack's ability to invert the target gradient. As shown in the left plot of Figure 7, larger client batch sizes lead to worse reconstruction quality. Smaller batches add variability to updates, making them more informative for the attacker, whereas larger batches smooth updates and reduce the amount of exploitable information. The right plot of Figure 7 shows that increasing the number of local iterations leads to worse reconstructions when the num-

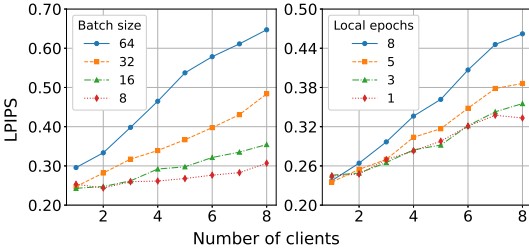

Figure 7: LPIPS of reconstructed images vs. number of clients, client batch size, and local iterations. Reconstruction quality worsens with more clients and larger batch sizes.

ber of clients is large. More local iterations cause greater gradient averaging, which dilutes the information needed to accurately reconstruct images from the target gradient. This is particularly important because federated learning often uses more local iterations to reduce communication.

**Unknown Learning Rate.** We assume in our experiments that the attacker knows the global learning rate $\eta_g$. This assumption simplifies the attack but need not be true for the attack to be effective. If the learning rate is applied globally (by the central server) and the attacker's guess differs from the true value, the target gradient will be inversely scaled by a factor of the ratio between the guessed learning rate and the true learning rate, leading to poor reconstruction quality. For simplicity, we set the base $\eta_g = 1$ and examine the impact on reconstruction when it is unknown to the attacker. The attacker uses its own training images to evaluate reconstruction quality as it knows they will be in the set of reconstructed images. Figure 8 shows that reconstruction quality degrades rapidly as the

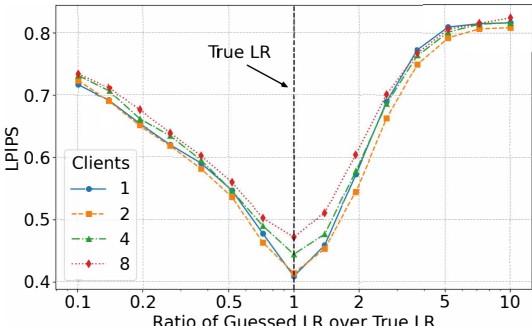

Figure 8: Quality of the raw reconstructed images when the attacker incorrectly guesses the global model's learning rate (LR). Image quality improves as the attacker guesses more correctly.

guessed and true learning rates diverge. However, within an order of magnitude of the true learning rate, the degradation follows a simple polynomial pattern. An attacker with sufficient computational resources can refine their guess over multiple iterations to improve reconstruction quality if the true learning rate is not known.

**Comparison to ROG and FedInverse.** We compare our attack method to FedInverse (Wu et al., 2024) and reconstruction from obfuscated gradients (ROG) (Yue et al., 2023). To the best of our knowledge, FedInverse is the only prior work addressing honest-but-curious client attacks. Unlike our approach, which reconstructs data from gradients, FedInverse inverts the global model. Their method performs best when the model is complex and trained over many epochs, whereas our attack works best with larger gradients, typically when the model is simpler and less trained. Figure 11 reveals that our method reconstructs higher-quality data from a small number of clients,

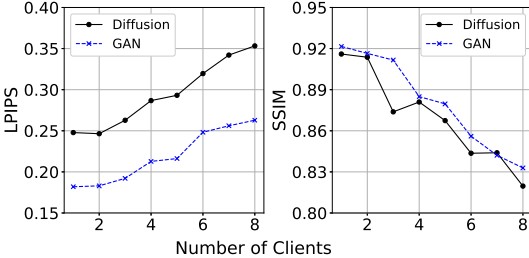

Figure 9: LPIPS and SSIM of reconstructed images using our direct postprocessor vs. GAN postprocessing with varying number of clients. Our method results in higher LPIPS as it blurs uncertain image details, compared to sharper but potentially less accurate outputs from GANs. SSIM, less sensitive to blurring, remains comparable across both approaches.

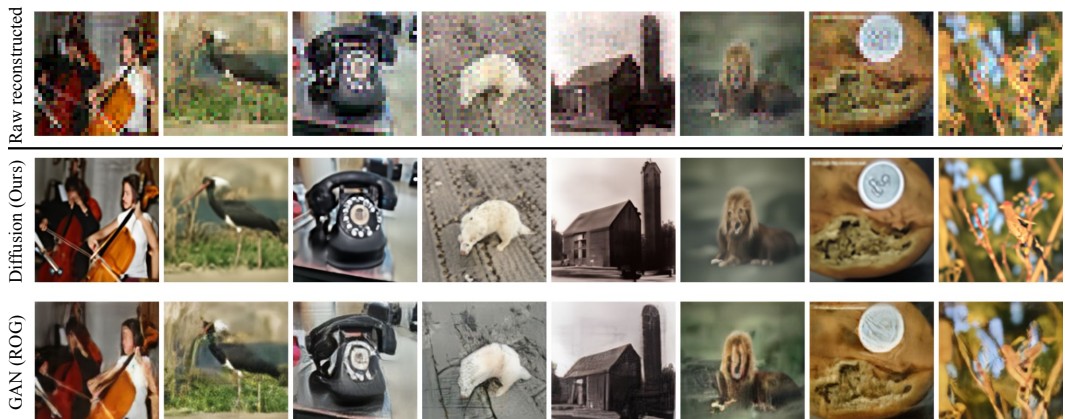

Figure 10: Pixel-level-correct reconstructed images using (row 2) our direct postprocessor and (row 3) ROG GAN (Yue et al., 2023). Our direct postprocessing method produces reconstructions that are more recognizable and have more accurate image details compared to the ROG GAN. However, our method also results in worse LPIPS and SSIM due to blurring.

whereas FedInverse produced lower-quality reconstructions from a larger number of clients. FedInverse is less sensitive to the number of clients, while our approach is more sensitive. Our proposed attack can be viewed as complementary to FedInverse.

We compare reconstruction quality from our direct and semantic postprocessing techniques to the state-of-the-art postprocessing results achieved by Yue et al. (2023). Figure 10 shows that our technique generates images that are more recognizable but often blurry because of uncertainty in the fine image details. LPIPS is designed to evaluate image quality, rather than detail accuracy, and penalizes blurriness and pixelation much more than hallucination. The left plot of Figure 9 shows that this results in the reconstructed images from our direct postprocessing technique having worse LPIPS than the state-of-the-art postprocessing technique. In contrast, the right plot of Figure 9 shows that our results achieve comparable SSIM, a metric that does not penalize blurriness as much as LPIPS.

## 5    CONCLUSIONS AND FUTURE WORK

We have demonstrated that a curious client attacker can successfully reconstruct high-quality images from a small number of clients simply by participating in two consecutive training rounds. This type of attack does not alter the training process or introduce corrupted data, making it difficult to detect by the server or other clients in the system. Our findings indicate that the attack is particularly effective when the number of participating clients is small or the available training examples are limited. This raises a significant concern for cross-silo FL, where participants collaborate specifically to overcome data scarcity (Li et al., 2020a). In such settings, our findings reveal a serious privacy risk, as the reconstruction of sensitive data becomes more feasible. Further research is needed to assess the robustness of more advanced FL frameworks against curious client attacks and develop effective defenses to preserve data privacy in cross-silo FL.

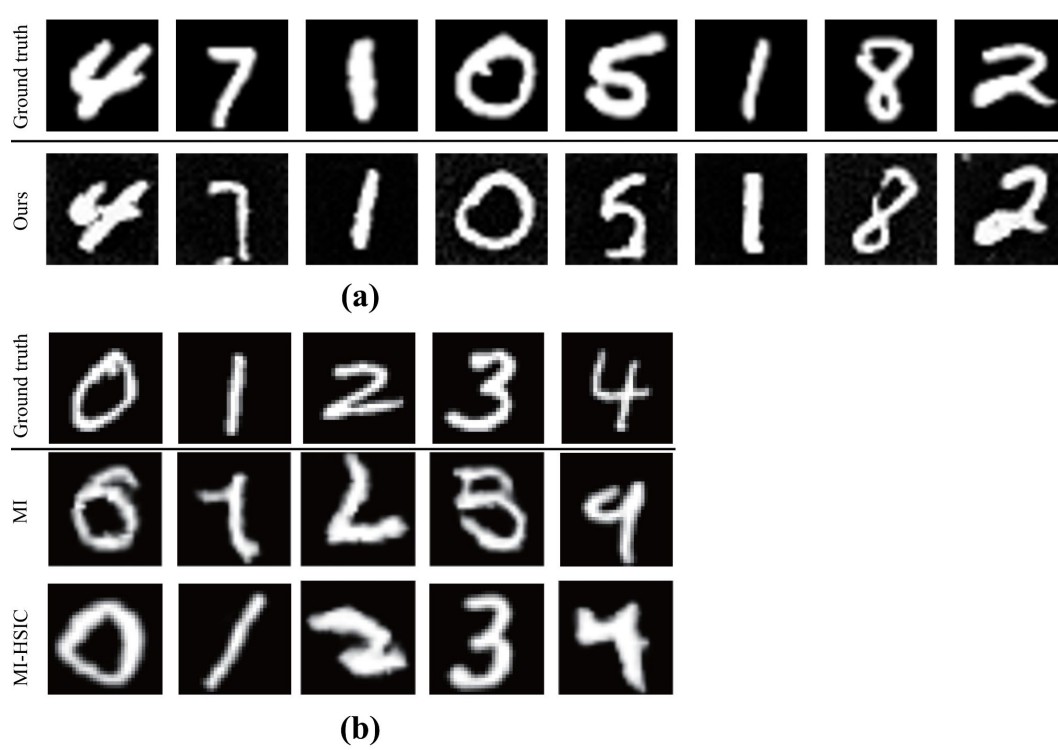

Figure 11: (a) Evaluation on MNIST of the attack with MDE postprocessing compared to (b) the model inversion (MI) and model inversion with Hilbert–Schmidt independence criterion (MI-HSIC) approaches [reproduced from Wu et al. (2024)]. Only 5 examples were provided for each method in Wu et al. 2024. Our reconstructed images are qualitatively more similar to the ground truth.

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

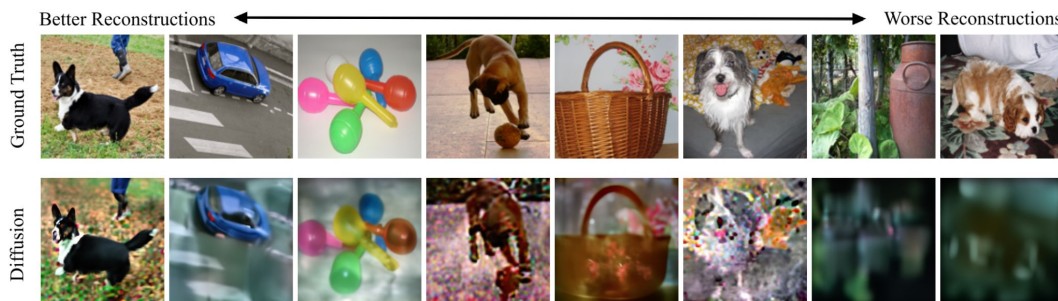

Figure 12: Pixel-level-correct reconstructed images from a system with 16 clients. With more clients, our reconstruction technique can reconstruct only a subset of training images with high quality, whereas others show distortion and color aliasing artifacts. Each client has 16 images and trains for 3 local iterations using LeNet5 as the model architecture.

## A  DISCUSSION

**Limitations.** Our attack struggles to reconstruct high-quality images in systems where the number of clients is large. As the information contained within the attacker's target gradient is averaged from more clients, it becomes more difficult to reconstruct high-quality images. With more than 8 clients, we observe that some reconstructed images remain high quality while others exhibit significant degradation or are not recognizable. Figure 12 shows the varying quality in our reconstructed images in a system of 16 clients. Additionally, we observe that the postprocessors are often able to restore image details that may not be obvious to a human observer looking at the raw reconstruction results. However, they are not able to restore images when the raw reconstruction result does not provide enough information, which is a problem common to all postprocessing tasks.

**Nonuniform Learning Rate.** The proposed attack relies on each client's gradient update being scaled by the same learning rate and this is also necessary for the global model to converge with FedAvg. To achieve the best model convergence during federated training, FedAvg scales each client's update by the client's number of training images, which gives the individual gradient of each training image equal weight in the global model update. If clients used very different learning rates, the clients with larger learning rates would dominate the global weight update, leading to suboptimal convergence. This is the basis for the assumption that the clients' learning rate in each round is either set by the server or otherwise controlled. For example, the clients may use a learning rate scheduler but agree on its parameters so the scale of their updates does not vary significantly in a given round. Regardless of how the learning rate is set during federated training, scaling individual image gradients unevenly in a way that would disrupt the attack is also likely to impede the global model's convergence.

**Unknown Number of Images.** The proposed attack assumes that the attacker can correctly guess the total number of training images $N$ in a given round. This assumption simplifies the attacking algorithm but is not always needed. Instead, the attacker can search for this integer value and decide on the best guess leading to a successful recovery of the attacker's own training images. If both the learning rate and number of images are unknown, a joint parameter search can be conducted.

**Application to Secure Aggregation.** We evaluate the similarity between our approach and server-side attacks against the secure aggregation protocol, identifying both significant differences and an additional application scenario of our attack. Our problem of inverting the aggregated gradients of multiple clients is similar to the problem server-side attackers encounter in systems using the secure aggregation protocol, which prevents the parameter server from knowing individual clients' gradients (Bonawitz et al., 2017). Despite this similarity, we have not found any other works that obtain high-quality reconstructions without modifying the global model (Shi et al., 2023; Zhao et al., 2024) or relying on additional information the server might have about the client devices, such as device type and available memory (Lam et al., 2021), which would not be possible for a client attacker. Most of these attacks also rely on information collected across many training rounds, which may not be available to a client who cannot choose which rounds it is selected to participate in. In contrast, our attack does not require the attacker to disrupt the training protocol or have

any information about the other clients beyond the model updates and total number of training images, which it may be able to guess. It also relies only on information from two consecutive training rounds. This indicates that our attack could also be performed by a server against a securely aggregated gradient and would allow it to avoid modifying the global model, maintaining the honest-but-curious threat model.

# B ASSUMPTIONS

Table 1: Many of the assumptions necessary for the proposed attack are shared by server-side gradient inversion attacks. We compare the assumptions necessary for our attack to ROG (Yue et al., 2023), DLG (Zhu et al., 2019), and iDLG (Zhao et al., 2020) to clarify which are unique to the curious-client threat model. Beyond what is required for these server-side attacks, the proposed attack requires that the number of clients in each training round is small and that the attacker knows or can guess the total number of images in a given round.

| Assumption | Ours (client) | ROG (server) | iDLG (server) | DLG (server) |
|---|---|---|---|---|
| Application: cross-silo/cross-device | cross-silo | both | both | both |
| Analytical label inversion | ✓ | ✓ | ✓ | |
| Single image per gradient | | | ✓ | |
| Each client trains on a single batch in each round | ✓ | ✓ | ✓ | ✓ |
| Clients can guess the total number of images in a given training round | ✓ | | | |
| Small number of clients | ✓ | | | |
| Small number of local iterations | ✓ | ✓ | ✓ | ✓ |
| Small number of images in each training round | ✓ | ✓ | ✓ | ✓ |
| Attacker has resources for complex attack | ✓ | ✓ | ✓ | ✓ |

## C EFFECT OF UNEVEN LOCAL BATCH SIZE

We compare the performance of the proposed attack with uneven client batch sizes to confirm that the proposed attack is not affected when training examples are distributed unevenly between clients. To evaluate this, we distribute a total of 256 training images unevenly across clients, using an average client batch size of 16 images. Half of the clients are initialized with 21 images (two-thirds of the total training data), while the other half receive 11 images (one-third of the total). Figure 13 compares the image reconstruction quality between this uneven distribution and a system where each client has an equal batch size of 16 images, keeping the total number of training images constant. The evaluation is conducted with an even number of clients ranging from 2 to 8. The results indicate negligible differences in reconstruction quality between the two systems. This finding supports our hypothesis that the weighting behavior of FedAvg renders the attack robust to uneven batch size distributions.

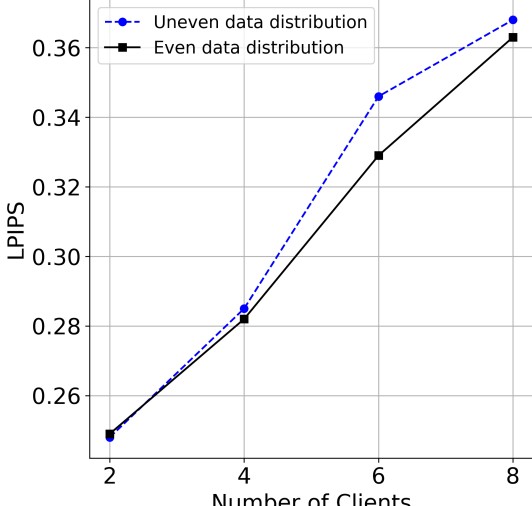

Figure 13: Quality of image reconstructions when training images are distributed unevenly between clients compared to an even distribution of training images. The effectiveness of the proposed attack is not sensitive to uneven client batch sizes, even when training for multiple local iterations.

## D EFFECT OF INVERSION LEARNING RATE

Figure 14 examines the sensitivity of the proposed attack to variations in the attacker's inversion learning rate, which is used optimize the dummy data. We evaluate reconstruction quality by varying both the inversion learning rate and the number of clients where the FL learning rate, used to update the global model, is fixed at 0.03. The results show only minor differences in image reconstruction quality across different learning rates, with slight variations in the optimal learning rate depending on the number of clients. Overall, the attack's performance remains robust as long as the inversion learning rate is within a reasonable range.

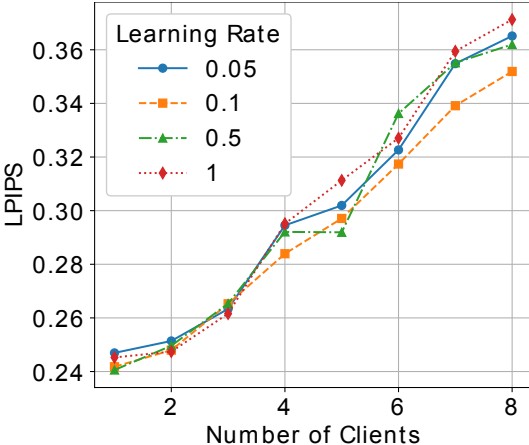

Figure 14: LPIPS of reconstructed images with varying attacker's inversion learning rate, which the reconstruction quality is not sensitive to.

## E   MORE RECONSTRUCTION RESULTS

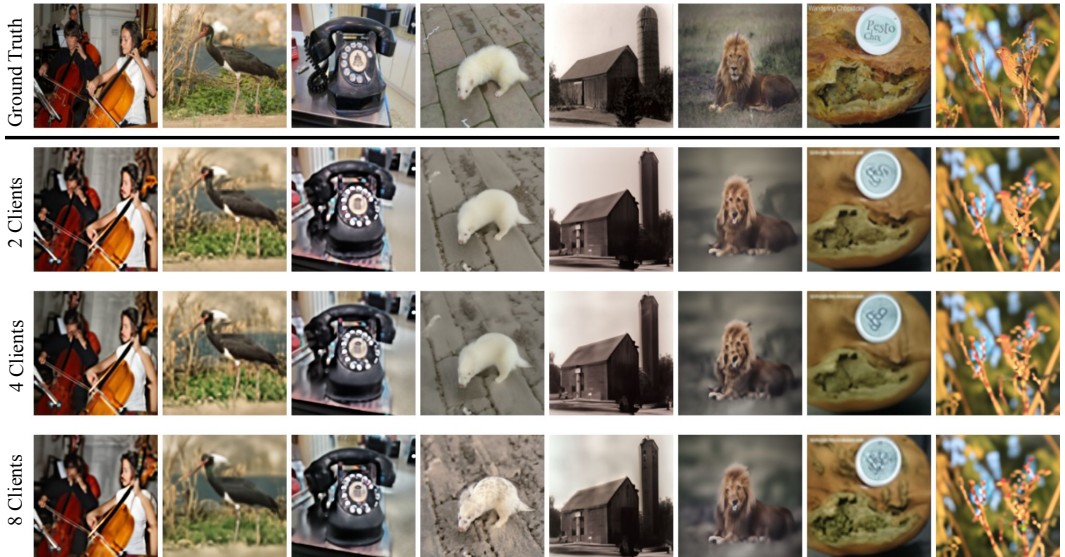

Figure 15: Images reconstructed by the proposed attack on a system with 8 images per client and three local iterations.

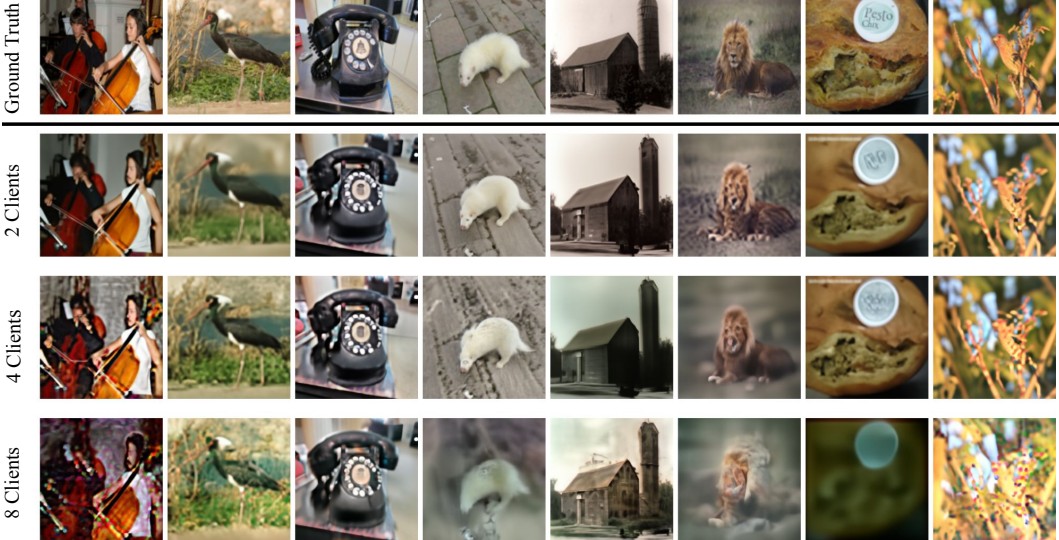

Figure 16: Images reconstructed by the proposed attack on a system with 32 images per client and three local iterations.

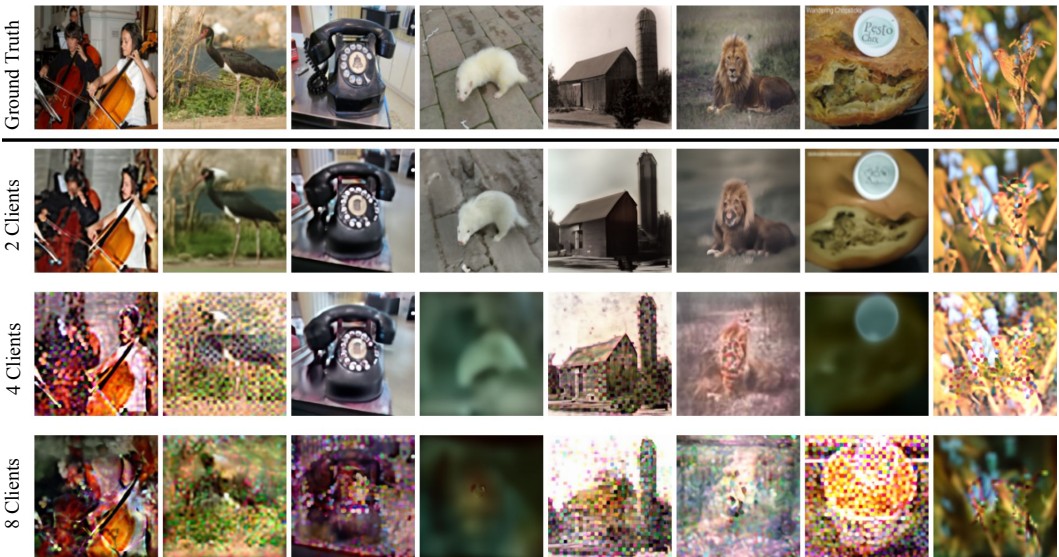

Figure 17: Images reconstructed by the proposed attack on a system with 64 images per client and three local iterations.

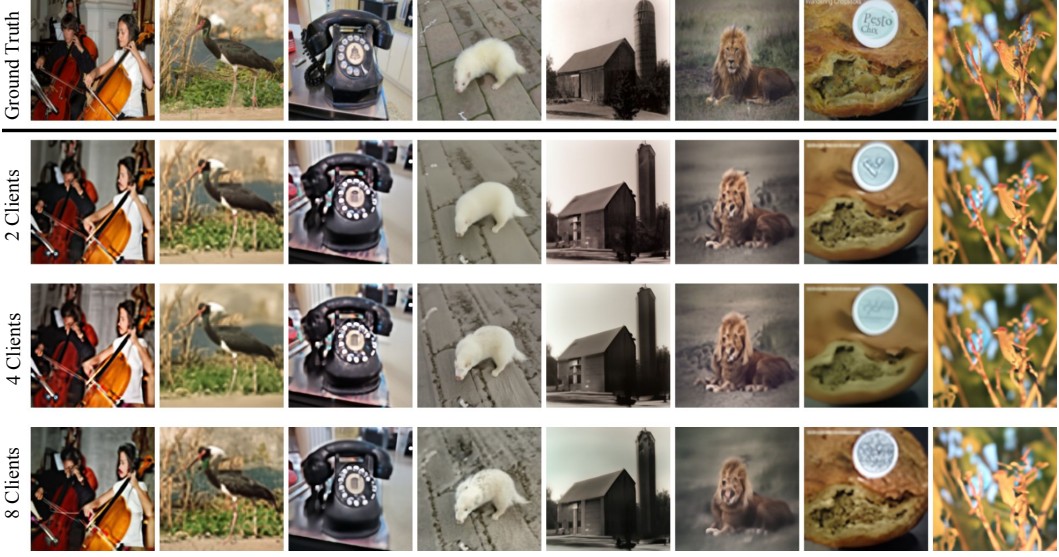

Figure 18: Images reconstructed by the proposed attack on a system with 16 images per client and one local iterations.

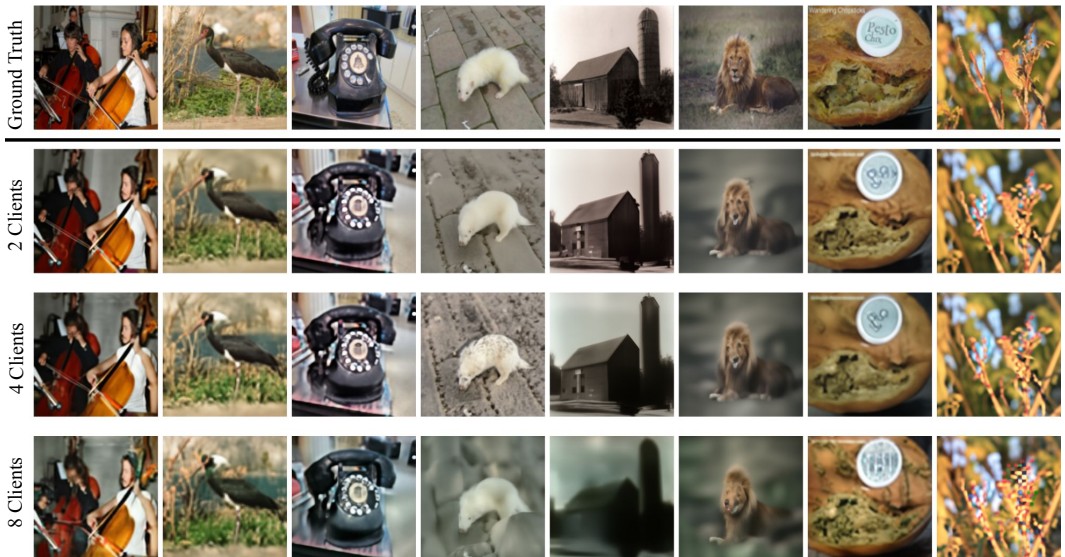

Figure 19: Images reconstructed by the proposed attack on a system with 16 images per client and five local iterations.

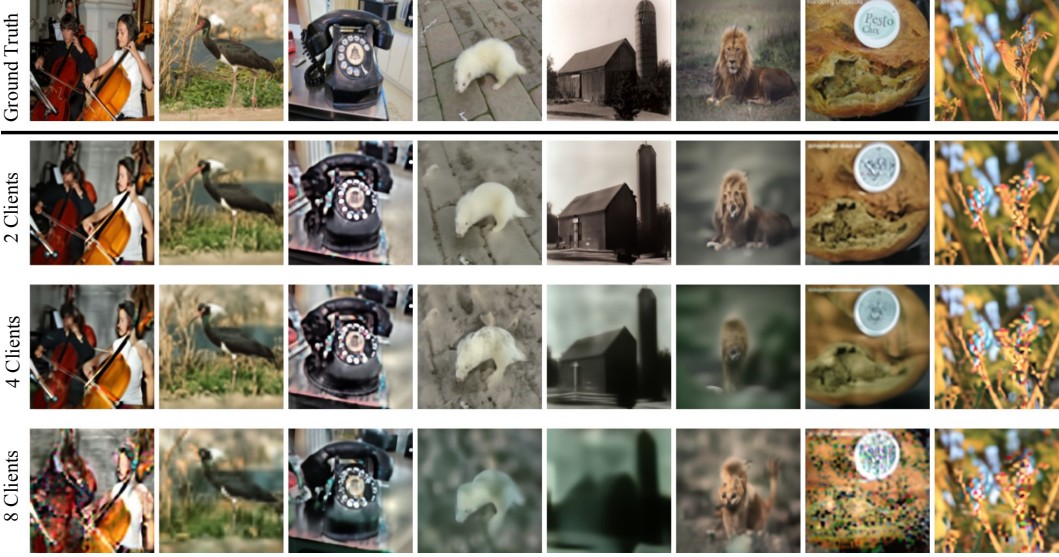

Figure 20: Images reconstructed by the proposed attack on a system with 16 images per client and eight local iterations.

