# OpenReview forum: "Federated Learning Nodes Can Reconstruct Peers' Image Data"
_ICLR.cc/2025/Conference — Submitted to ICLR 2025_

### Official Review · Reviewer_i2hB · 2024-10-24

**Soundness:** 2
**Presentation:** 2
**Contribution:** 1
**Rating:** 3
**Confidence:** 5

**Summary:**

This paper proposes a high-quality privacy data reconstruction method in the federated learning scenario, which achieves excellent results especially when the number of participating training nodes and the amount of data are limited. Unlike traditional methods, this paper considers attacks from peer nodes, making the scenario more versatile.

**Strengths:**

This paper integrates gradient inversion attacks with generative models to achieve higher quality privacy attacks. Additionally, it takes into account attacks from peer nodes, making the scenario more versatile compared to traditional ones.

**Weaknesses:**

The authors claim that the advantage of this paper lies in the achievement of node-level privacy attacks in the federated learning scenario. However, there are several significant limitations:
1. Unreasonable assumptions. In the aggregation of global gradients, updates from different nodes are weighted based on the amount of training data used by each party. However, the authors simplistically assume that the parties aggregate with equal weights. Additionally, the authors mention that an attacker can directly initialize N (line 161), but we do not think that a peer node can have this information. In summary, these assumptions make the paper fundamentally similar to traditional privacy attacks conducted by a central server, with only an additional simple subtraction operation. This also makes the paper’s main contribution less solid.

2. Lack of novelty and originality. The work presented in this paper is a simple combination of gradient inversion attacks with super-resolution and denoising techniques, without proposing a new method or solving an unsolved problem (although the authors claim to have achieved peer node attacks, we have already shown in point 1 that this assumption is not fundamentally different from existing center-based studies). Therefore, the academic value of this paper is not sufficient for publication in a top-tier conference like ICLR.

3. The introduction of the experimental setup is very rough. In the optimization objective Eq. (3) of this paper, there is no variable related to labels, and the setup does not explain how the attacker obtains the labels. Although the paper mentions the method of Yin et al., their method cannot handle scenarios with duplicate labels. Moreover, I do not think that the paper can be replicated based on the setup provided, as almost all configurations related to optimization are missing.

**Questions:**

My concerns are detailed in the WEAKNESS part, specifically in relation to these limitations. If the authors provide convincing responses to these limitations, I would be happy to raise my score.

---

> ### Author Response · Authors · 2024-11-28
> **Response to Reviewer i2hB (Part 1)**
>
> **Comment 1 (Reviewer i2hB)** Unreasonable assumptions. In the aggregation of global gradients, updates from different nodes are weighted based on the amount of training data used by each party. However, the authors simplistically assume that the parties aggregate with equal weights. Additionally, the authors mention that an attacker can directly initialize N (line 161), but we do not think that a peer node can have this information. In summary, these assumptions make the paper fundamentally similar to traditional privacy attacks conducted by a central server, with only an additional simple subtraction operation. This also makes the paper’s main contribution less solid.
>
> **Response:** We thank the reviewer for their comments. We address the comments below on the assumption that clients aggregate with equal weights and that the attacker knows $N$.
>
> First, we appreciate the reviewer for pointing out that FedAvg should weight each client's update based on the amount of training data. We have updated Section 3.1 to account for weighting based on each client's batch size $N_k$. We note that this weighting mechanism leads to each image's individual gradient contributing equally to the global model update and therefore, allows us to remove the assumption of uniform local batch size.
>
> Second, we now assume that the attacker can correctly guess the total number of images $N$. We have incorporated this assumption into our threat model (Section 3.1) of the revised manuscript, which reads
>
> >"The attacker may not know the number of clients in each training round but can correctly guess the total number of training images."
>
> We have also added a discussion on how the attacker may estimate the total number of training images $N$ in Section 4 of the updated manuscript. We quote it as follows:
>
> >"The proposed attack assumes that the attacker can correctly guess the total number of training images $N$ in a given round. This assumption simplifies the attacking algorithm but is not always needed. Instead, the attacker can search for this integer value and decide on the best guess leading to a successful recovery of the attacker's own training images. If both the learning rate and number of images are unknown, a joint parameter search can be conducted."
>
>
> **Comment 2 (Reviewer i2hB):** Lack of novelty and originality. The work presented in this paper is a simple combination of gradient inversion attacks with super-resolution and denoising techniques, without proposing a new method or solving an unsolved problem (although the authors claim to have achieved peer node attacks, we have already shown in point 1 that this assumption is not fundamentally different from existing center-based studies). Therefore, the academic value of this paper is not sufficient for publication in a top-tier conference like ICLR.
>
> **Response:** We are among the first to identify federated training’s vulnerability posed by honest-but-curious clients and the first to demonstrate that such attackers can reconstruct high-quality data. In addition, our inversion framework with diffusion models is able to reconstruct image data from diluted gradients of clients without the need to know the number of clients and the image counts for all clients.
>
> The curious-client inversion attack having to deal with $K$ client models is a more difficult gradient inversion problem than curious-server inversion attackers that have to deal with only one client model, because the number of clients and the image counts for all clients are usually unknown. Our optimization algorithm addresses this difficulty through using a single model $\mathbf{W}^{(t,u)}$ to capture the net effect of $K$ clients’ local models $\mathbf{W}_k^{(t,u)}$. A detailed description of our optimization algorithm is in Section 3 of the revised manuscript:
>
> >“The attacker passes them through a global model and compares the resulting gradient update $\Delta^{(t)}(\mathbf{X}, \mathbf{Y}) = \sum_{u=0}^{\tau-1}\sum_{l=1}^{N} \nabla\ell(\mathbf{W}^{(t,u)}; \mathbf{X}\_{l}; \mathbf{Y}\_{l})$ to the target gradient … the evolving global model $\\{\mathbf{W}^{(t,u)}\\}_{u=0}^{\tau-1}$ requires only the knowledge of the total number of images, eliminating the need to know the number of clients and the image counts from all clients.”
>
>
> **Comment 3 (Reviewer i2hB):** In the optimization objective Eq. (3) of this paper, there is no variable related to labels.
>
> **Response:**  We thank the reviewer for pointing out the mistake of neglecting the labels variable. We have revised our equations in Section 3.1 to include a variable for labels.

---

> > ### Comment · Reviewer_i2hB · 2024-11-28
> >
> > I'm sorry that you are not the first to identify federated training’s vulnerability posed by honest-but-curious clients.
> > Please refer to Part D, Section 2 in paper [1]  (especially Eq. (5) and Fig. 2).
> >
> > [1] Li, Zhaohua, et al. "A survey of image gradient inversion against federated learning." Authorea Preprints (2023).
> > url: https://www.techrxiv.org/doi/pdf/10.36227/techrxiv.18254723.v1

---

> > > ### Author Response · Authors · 2024-11-30
> > >
> > > We thank the reviewer for pointing out that Li et al. (2023) discussed the honest-but-curious threat model in their survey paper. We note that even though the FL literature (Kairouz et al., 2021; Li et al., 2023) has mentioned the honest-but-curious client threat model on multiple occasions, to the best of our knowledge, our paper is the first to experimentally demonstrate this threat on multiple ML datasets. We will update our manuscript to reflect the fact that our contribution focuses on the experimental evaluation and not on proposing the notion of honest-but-curious vulnerability. If the reviewer is aware of any other works that have demonstrated successful reconstruction attacks under this threat model, we are happy to compare them to our attack.
> > >
> > >
> > > Reference:
> > >
> > > Peter Kairouz, H Brendan McMahan, Brendan Avent, AurÅLelien Bellet, Mehdi Bennis, Arjun Nitin Bhagoji, Kallista Bonawitz, Zachary Charles, Graham Cormode, Rachel Cummings, et al. “Advances and open problems in federated learning.” Foundations and Trends. in Machine Learning, 14(1–2):1–210, 2021.

---

> ### Author Response · Authors · 2024-11-28
> **Response to Reviewer i2hB (Part 2)**
>
> **Comment 4 (Reviewer i2hB):** The setup does not explain how the attacker obtains the labels. Although the paper mentions the method of Yin et al., their method cannot handle scenarios with duplicate labels.
>
> **Response:**  We have updated our threat model in Section 3.1 to specify that we assume a successful label recovery based on Ma et al. (2023), which can handle scenarios with duplicate labels. The updated sentence reads
> >"...  the class labels have been analytically inverted as in Ma et al. (2023). …"
>
> **Comment 5 (Reviewer i2hB):** The introduction of the experimental setup is very rough. I do not think that the paper can be replicated based on the setup provided, as almost all configurations related to optimization are missing.
>
> **Response:** We have updated our experimental conditions to include the parameters we use for FL learning rate, attacker learning rate, number of local iterations, and the scale factor for bicubic downsampling of the dummy data. Our experimental setup is complete based on the revised second paragraph of Section 4, which reads
>
> >" … Each client performs 3 iterations of local training on 16 images as this batch size provides a baseline where the attack reconstructs recognizable images from the target gradient… The attacker uses a learning rate of 0.1 to optimize the dummy data and the attack is conducted after the first FL round, following the approach of Yue et al. (2023)... Before inverting the target gradient, the attacker encodes its dummy data through bicubic sampling with a scale factor of 4 to reduce the number of unknown parameters."

---

### Official Review · Reviewer_kGL7 · 2024-10-30

**Soundness:** 3
**Presentation:** 3
**Contribution:** 2
**Rating:** 5
**Confidence:** 5

**Summary:**

This paper investigates the privacy issues in federated learning (FL), a framework allowing nodes to train models locally while sharing updates. Despite its privacy goals, FL is vulnerable to data reconstruction attacks. The paper reveals that not only central servers but also semi-honest clients can reconstruct peers' image data, posing significant risks. Using advanced diffusion models, the authors show how a single client can enhance image quality, underscoring the need for stronger privacy measures to prevent client-side attacks in FL.

**Strengths:**

1. **Stealth and Undetectability**: The attack method does not disrupt the training process or introduce corrupted data, making it challenging for detection by servers or other clients, which underscores its potential impact.

2. **Relevance to Cross-Silo FL**: The findings are particularly concerning for cross-silo FL, where data scarcity is addressed through collaboration, emphasizing the need for enhanced privacy measures in such settings.

3. **Extensive Experiments**: The paper conducts thorough experiments to validate the effectiveness of the attack, providing strong empirical evidence of the vulnerability in FL systems.

**Weaknesses:**

1. This paper attacks from the perspective of any node/client and reconstruct all training data of all other participants. However, this is no different from a conventional inversion attack launched from the server.
When the secure aggregation protocol is applied, the server can obtain the model parameters at time $t$ and the corresponding aggregated gradients; while any client can receive the model parameters at time $t$ and time $t+1$. Obviously, the information obtained in these two cases is exactly the same, and the updated gradient is the difference between these two rounds.
It is good that the authors start from the node/client perspective, but the current analysis is the same as the typical gradient inversion attacks, and there is no special or new inspiration.

2. The core contribution of the paper is to propose a post-processing method for reconstructing images (based on the diffusion models). However, this is based on a premise that the original restored image already contains enough information.
If the results after the attack are like the results of Figure 5(b) and Figure 15 of ROG or the last three rows of Figure 4 of GradInversion (See through Gradients, Yin et al.), the reconstructed images are similar to noise, then your proposed method obviously does not work. How do you solve this situation? This is not mentioned in the paper.

**Questions:**

1. Figures 2, 3, 5, and 8 demonstrate the effect of data reconstruction. What hyperparameters such as the training model structure, batch size, and epoch of FedAvg local training are used corresponding to these results?

2. You selected LPIPS as the main evaluation metric. Do the results or trends of MSE, PSNR, SSIM and LPIPS are consistent in these experiments? Because sometimes the LPIPS values ​​of two sets of images may be close, but the visual effects are very different.

3. In the left figure of Figure 6, when there are 8 clients and the batch size is 64, the attacker has to restore a total of (amazing) 512 images. What is the specific visualizatioin results of these images? Does the LPIPS value reflect the actual reconstruction effect?

4. In Equation (3), why do you choose L2 norm instead of cosine similarity? In your method, which one do you think has a greater impact on the final restoration result, raw reconstruction or post-processing?

5. Figure 2 shows the results of different epochs. How do you choose the best epoch? For all the images to be processed, do they use the same optimal number of epochs?

6. After adding two diffusion models (MDT and DDPMs) to optimize the reconstruction results, how much will the efficiency of the attack and the computational cost increase compared to before?

---

> ### Author Response · Authors · 2024-11-28
> **Response to Reviewer kGL7 (Part 1)**
>
> **Comment 1 (Reviewer kGL7):** Figures 2, 3, 5, and 8 demonstrate the effect of data reconstruction. What hyperparameters such as the training model structure, batch size, and epoch of FedAvg local training are used corresponding to these results?
>
> **Response:** We thank the reviewer for their comments. We have updated the captions for Figures 5 and 8 to specify that each uses 16 images per client, trains for 3 local epochs, and uses LeNet5 as the model architecture. The results shown in Figures 2 and 3 use the same hyperparameters but we have chosen not to detail information about the raw attack parameters as those figures are intended to highlight how the postprocessing modules can visually enhance the raw reconstructed images.
>
> Additionally, we have updated our experimental conditions section to specify that our attack is performed after the first FL round. This approach is commonly used in the curious-server literature, such as by ROG (Yue et al., 2023). The updated section reads
>
> >"... The attack was conducted after the first FL round, following the approach of Yue et al., (2023). … "
>
> Quantifying how the reconstruction quality is affected by attacking later rounds of model training is a promising direction for future work.
>
> **Comment 2 (Reviewer kGL7)** You selected LPIPS as the main evaluation metric. Do the results or trends of MSE, PSNR, SSIM and LPIPS are consistent in these experiments? Because sometimes the LPIPS values ​​of two sets of images may be close, but the visual effects are very different.
>
> **Response:** Yes, we observe similar trends in SSIM and PSNR/MSE. We have updated our experimental conditions to clarify our choice of image quality metrics. The updated section reads
>
> >"We use LPIPS (Zhang et al., 2018) as the primary metric to evaluate the quality of the attack's reconstructed images as it provides the best representation of perceptual image quality based on our experiments, though we observe similar trends for SSIM (Wang et al., 2004) and PSNR/MSE."
>
> **Comment 3 (Reviewer kGL7)** In the left figure of Figure 6, when there are 8 clients and the batch size is 64, the attacker has to restore a total of (amazing) 512 images. What is the specific visualization results of these images? Does the LPIPS value reflect the actual reconstruction effect?
>
> **Response:** We have added more reconstruction results in newly added Appendix C that show a random sampling of the reconstructed images with varying hyperparameters, including the case with local batch size 64 and 8 clients, which is shown in the bottom row of the newly added Figure 16.
>
> **Comment 4 (Reviewer kGL7):** In Equation (3), why do you choose L2 norm instead of cosine similarity?
>
> **Response:** We followed the standard practice in the literature such as ROG (Yue et al., 2023). We did not encounter any issues with the L2 norm and the L2 norm produced better results than using cosine similarity.
>
> **Comment 5 (Reviewer kGL7):** Figure 2 shows the results of different epochs. How do you choose the best epoch? For all the images to be processed, do they use the same optimal number of epochs?
>
> **Response:** For the semantic postprocessor, our main results section details how the optimal epoch was chosen:
>
> >"We observe that at an optimal epoch number, the output images closely match the target, preserving its geometric structure and perceptual features with photorealistic quality. This optimal point varies across target images and was determined qualitatively."
>
> We have made a minor modification to this section to specify that we do not use the ground truth image to select the best epoch as this would be unrealistic for real-world applications where the ground truth is unknown. The updated sentence reads
>
> >"This optimal point varies across target images and was determined qualitatively based on the raw reconstructed images."
>
> **Comment 6 (Reviewer kGL7):** After adding two diffusion models (MDT and DDPMs) to optimize the reconstruction results, how much will the efficiency of the attack and the computational cost increase compared to before?
>
> **Response:** Our proposed postprocessing methods do increase the attack overhead significantly. This issue is common when using diffusion models as they require iterative refinement of each batch of images. We do not evaluate the computational efficiency of the attack based on our assumption that the attacker has sufficient computational resources in cross-silo scenarios that this paper assumes.

---

> ### Author Response · Authors · 2024-11-28
> **Response to Reviewer kGL7 (Part 2)**
>
> **Comment 7 (Reviewer kGL7):** This paper attacks from the perspective of any node/client and reconstruct all training data of all other participants. However, this is no different from a conventional inversion attack launched from the server. When the secure aggregation protocol is applied, the server can obtain the model parameters at time?and the corresponding aggregated gradients; while any client can receive the model parameters at time t and time t+1. Obviously, the information obtained in these two cases is exactly the same, and the updated gradient is the difference between these two rounds. It is good that the authors start from the node/client perspective, but the current analysis is the same as the typical gradient inversion attacks, and there is no special or new inspiration.
>
> **Response:** In response to the reviewer's comment, we have conducted an additional literature review of curious-server attacks on the secure aggregation protocol. We have included a discussion in Appendix A of the revised draft, which reads
>
> >"We evaluate the similarity between our approach and server-side attacks against the secure aggregation protocol, identifying both significant differences and an additional application scenario of our attack. Our problem of inverting the aggregated gradients of multiple clients is similar to the problem server-side attackers encounter in systems using the secure aggregation protocol, which prevents the parameter server from knowing individual clients' gradients (Bonawitz et al., 2017). Despite this similarity, we have not found any other works that obtain high-quality reconstructions without modifying the global model (Shi et al., 2023; Zhao et al., 2024) or relying on additional information the server might have about the client devices, such as device type and available memory (Lam et al., 2021), which would not be possible for a client attacker. Most of these attacks also rely on information collected across many training rounds, which may not be available to a client who cannot choose which rounds it is selected to participate in. In contrast, our attack does not require the attacker to disrupt the training protocol or have any information about the other clients beyond the model updates and total number of training images, which it may be able to guess. It also relies only on information from two consecutive training rounds. This indicates that our attack could also be performed by a server against a securely aggregated gradient and would allow it to avoid modifying the global model, maintaining the honest-but-curious threat model."
>
>
> **Comment 8 (Reviewer kGL7):** The core contribution of the paper is to propose a post-processing method for reconstructing images (based on the diffusion models). However, this is based on a premise that the original restored image already contains enough information. If the results after the attack are like the results of Figure 5(b) and Figure 15 of ROG or the last three rows of Figure 4 of GradInversion (See through Gradients, Yin et al.), the reconstructed images are similar to noise, then your proposed method obviously does not work. How do you solve this situation? This is not mentioned in the paper.
>
> **Response:** We have added an additional discussion on the postprocessors to Appendix A of the revised draft, which reads
>
> >"We observe that the postprocessors are often able to restore image details that may not be obvious to a human observer looking at the raw reconstruction results. However, they are not able to restore images when the raw reconstruction result does not provide enough information, which is a problem common to all postprocessing tasks."
>
> **Comment 9 (Reviewer kGL7)** In your method, which one do you think has a greater impact on the final restoration result, raw reconstruction or post-processing?
>
> **Response:** Given that the postprocessing modules are unable to reconstruct images when the raw attack does not provide enough information about their content, the raw attack tends to have a stronger fundamental impact on the overall reconstruction quality.

---

### Official Review · Reviewer_YKmd · 2024-11-03

**Soundness:** 2
**Presentation:** 3
**Contribution:** 2
**Rating:** 3
**Confidence:** 4

**Summary:**

This paper proposes an attack approach within the federated learning (FL) framework to reconstruct image data from participating peers in a centralized system. The study demonstrates that consecutive updates in the FL setting can inadvertently reveal information about other clients. Experiments are conducted to validate the effectiveness of this attack method.

**Strengths:**

The strength of this paper lies in its successful implementation of an attack capable of reconstructing images from other participating users. The experiments effectively demonstrate the effectiveness of the proposed method.

**Weaknesses:**

I have several concerns regarding the experimental setup and the novelty of this paper, which I outline below:

The method relies on several assumptions, including that each client has ample computational resources, employs a consistent learning rate, and trains with an equal number of images locally. Additionally, the approach presumes that the attacker is either aware of or can accurately estimate the number of clients participating in each training round. Further assumptions, such as the use of full-batch gradient descent for local training and other idealized conditions, may not be realistic for federated learning (FL) environments.

These restrictive assumptions may limit the method's practical applicability in typical FL settings. Federated learning is generally designed to support users with limited resources, accommodate non-iid (non-independent and identically distributed) data, and handle asynchronous updates among clients. The paper does not address these critical FL challenges, potentially reducing the relevance of the proposed approach in real-world scenarios.

The optimization framework introduced here does not appear to be novel, and the paper lacks citations to previous work on similar frameworks. There is also no comparison provided to demonstrate why or how the proposed optimization function is more effective or advantageous over existing methods.

In the experiments, the maximum number of clients is set at 8, which limits the insights into how the framework performs at larger scales. Additionally, there are insufficient ablation studies to illustrate the robustness of the proposed framework under varying conditions.

**Questions:**

How does the proposed approach perform with a larger number of clients?
How robust is the proposed approach if any of its underlying assumptions are violated?

---

> ### Author Response · Authors · 2024-11-28
> **Response to Reviewer Ykmd (Part 1)**
>
> **Comment 1 (Reviewer Ykmd):** The method relies on several assumptions, including that each client has ample computational resources, employs a consistent learning rate, and trains with an equal number of images locally. Additionally, the approach presumes that the attacker is either aware of or can accurately estimate the number of clients participating in each training round. Further assumptions, such as the use of full-batch gradient descent for local training and other idealized conditions, may not be realistic for federated learning (FL) environments. These restrictive assumptions may limit the method's practical applicability in typical FL settings.
> How robust is the proposed approach if any of its underlying assumptions are violated?
>
> **Response:** We thank the reviewer for their comments. Regarding the main assumptions made by our work, we have performed additional experiments based on the reviewer's comments and were able to remove the assumption of uniform batch size. This result also allows us to remove the assumption that the attacker knows the number of clients if it can guess the total number of images used for the training round. We further show that our assumptions of uniform learning rate and synchronous gradient updates are appropriate for our use case of cross-silo federated learning. We also note that the assumption that each client performs full-batch gradient descent during a single training round is shared by other related works. We now discuss each assumption in detail.
>
> **Uniform batch size.** We conducted extra experiments during the rebuttal period and have been able to show that the proposed attack is not affected if clients have different local batch sizes. Intuitively, this is because FedAvg scales each client's update by the number of training images, which leads to each image's individual contribution being weighted equally. We have updated Section 3.1 of the manuscript to account for this scaling behavior. The updated section reads,
>
> > "... The server generates the global weights by a weighted average of all clients' final local weights …  we obtain the global weight update equation:
> \begin{equation}
> \mathbf{W}^{(t+1)} = \mathbf{W}^{(t)} - \frac{\eta_{\text{g}}}{N} \Delta_k.
> \end{equation}
> … We note that scaling each client's update by its number of training images $N_k$ causes the gradient of each training image to be weighted equally in the global update."
>
>
> We have also performed additional experiments to confirm that the attack performance is not sensitive to uneven client training batches. The outcome is included in newly added Appendix C of the revised manuscript, which reads
>
> >"We compare the performance of the proposed attack with uneven client batch sizes to confirm that the proposed attack is not affected when training examples are distributed unevenly between clients. To evaluate this, we distribute a total of 256 training images unevenly across clients, using an average client batch size of 16 images. Half of the clients are initialized with 21 images (two-thirds of the total training data), while the other half receive 11 images (one-third of the total). Figure 13 compares the image reconstruction quality between this uneven distribution and a system where each client has an equal batch size of 16 images, keeping the total number of training images constant. The evaluation is conducted with an even number of clients ranging from 2 to 8. The results indicate negligible differences in reconstruction quality between the two systems. This finding supports our hypothesis that the weighting behavior of FedAvg renders the attack robust to uneven batch size distributions."
>
> **Known number of clients.** With our revised assumptions on batch size and learning rate, it is now evident that the attacker does not need to know the number of clients if it can guess the total number of training images. Given that each image's gradient update is weighted equally by FedAvg, the only information the attacker would need to know or guess to initialize the dummy data batch size is the total number of images used in the first of the two consecutive training rounds. Our updated assumption in Section 3.1 of the manuscript reads
>
> >"The attacker may not know the number of clients in each training round but can correctly guess the total number of training images."
>
> Additionally, we have added a discussion on how the attacker could determine this number in Appendix A of the updated manuscript, which reads
>
> >"The proposed attack assumes that the attacker can correctly guess the total number of training images $N$ in a given round. This assumption simplifies the attacking algorithm but is not always needed. Instead, the attacker can search for this integer value and decide on the best guess leading to a successful recovery of the attacker's own training images. If both the learning rate and number of images are unknown, a joint parameter search can be conducted."

---

> ### Author Response · Authors · 2024-11-28
> **Response to Reviewer Ykmd (Part 2)**
>
> **Learning Rate.** We have added an additional discussion to Appendix A of the updated manuscript to support our assumption that each client's gradient update is scaled by the same learning rate. The updated section reads
>
> >"The proposed attack relies on each client's gradient update being scaled by the same learning rate but this is also necessary for the global model to converge with FedAvg. To achieve the best model convergence during federated training, FedAvg scales each client's update by the client's number of training images, which gives the individual gradient of each training image equal weight in the global model update. If clients used very different learning rates, the clients with larger learning rates would dominate the global weight update, leading to suboptimal convergence. This is the basis for the assumption that the clients' learning rate in each round is either set by the server or otherwise controlled. For example, the clients may use a learning rate scheduler but agree on its parameters so the scale of their updates does not vary significantly in a given round. Regardless of how the learning rate is set during federated training, scaling individual image gradients unevenly in a way that would disrupt the attack is also likely to impede the global model's convergence."
>
> **Full-batch gradient descent.** While this assumption is simplifying, it is not unique to our work. As shown in the newly added Table 1 of Appendix B, prior works such as ROG and DLG assume this in the implementation without stating it explicitly.
>
> | **Assumption**                                     | **Ours** (client) | **ROG** (server) | **iDLG** (server) | **DLG** (server) |
> |----------------------------------------------------|-------------------|------------------|-------------------|------------------|
> | Application: cross-silo/cross-device              | cross-silo        | both             | both              | both             |
> | Analytical label inversion                        | ✓                 | ✓                | ✓                 |                  |
> | Single image per gradient                         |                   |                  | ✓                 |                  |
> | Each client trains on a single batch in each round| ✓                 | ✓                | ✓                 | ✓                |
> | Clients can guess the total number of images in a given training round | ✓ |                  |                   |                  |
> | Small number of clients                           | ✓                 |                  |                   |                  |
> | Small number of local iterations                  | ✓                 | ✓                | ✓                 | ✓                |
> | Small number of images in each training round     | ✓                 | ✓                | ✓                 | ✓                |
> | Attacker has resources for complex attack         | ✓                 | ✓                | ✓                 | ✓                |
>
> **Comment 2 (Reviewer Ykmd):** Federated learning is generally designed to support users with limited resources, accommodate non-iid (non-independent and identically distributed) data, and handle asynchronous updates among clients. The paper does not address these critical FL challenges, potentially reducing the relevance of the proposed approach in real-world scenarios.
>
> **Response:**  We thank the reviewer for their assessment. We note that our work targets cross-silo, rather than cross-device federated learning as the reviewer was implying in their comment. We have updated our threat model in Section 3.1 of the manuscript to clarify this point. The updated section reads,
>
> >"We target cross-silo FL scenarios, in which a small number of clients collaborate to overcome data scarcity. For example, a group of hospitals may use FL to develop a classifier for rare diseases from CT scans, where each has limited training examples and images cannot be directly shared due to privacy concerns. We assume that the system is designed to prioritize model accuracy and uses synchronous gradient updates. Clients are not edge devices and have sufficient computational resources to perform the optimization process while participating in FL."

---

> ### Author Response · Authors · 2024-11-28
> **Response to Reviewer Ykmd (Part 3)**
>
> **Comment 3 (Reviewer Ykmd):** The optimization framework introduced here does not appear to be novel, and the paper lacks citations to previous work on similar frameworks. There is also no comparison provided to demonstrate why or how the proposed optimization function is more effective or advantageous over existing methods.
>
> **Response:**  Our inversion framework with diffusion models is able to reconstruct image data from diluted gradients of clients without the need to know the number of clients and the image counts for all clients.
>
> The curious-client inversion attack having to deal with $K$ client models is a more difficult gradient inversion problem than curious-server inversion attackers that have to deal with only one client model, because the number of clients and the image counts for all clients are usually unknown. Our optimization algorithm addresses this difficulty through using a single model $\mathbf{W}^{(t,u)}$ to capture the net effect of $K$ clients’ local models $\mathbf{W}_k^{(t,u)}$. A detailed description of our optimization algorithm is in Section 3 of the revised manuscript:
>
> >“The attacker passes them through a global model and compares the resulting gradient update $\Delta^{(t)}(\mathbf{X}, \mathbf{Y}) = \sum_{u=0}^{\tau-1}\sum_{l=1}^{N} \nabla\ell(\mathbf{W}^{(t,u)}; \mathbf{X}\_{l}; \mathbf{Y}\_{l})$ to the target gradient … the evolving global model $\\{\mathbf{W}^{(t,u)}\\}_{u=0}^{\tau-1}$ requires only the knowledge of the total number of images, eliminating the need to know the number of clients and the image counts from all clients.”
>
> To the best of our knowledge, the paper has cited relevant work on server-side and client-side attacks. If the reviewer is willing to provide other references, we are happy to include them in the manuscript.
>
>
> **Comment 4 (Reviewer Ykmd):** There are insufficient ablation studies to illustrate the robustness of the proposed framework under varying conditions.
>
> **Response:** We have performed additional experiments to demonstrate that the approach is robust to uneven client batch size, which are outlined in our response to Comment 1 and included in the newly added Appendix B of the revised draft.
>
> **Comment 5 (Reviewer Ykmd):** In the experiments, the maximum number of clients is set at 8, which limits the insights into how the framework performs at larger scales. How does the proposed approach perform with a larger number of clients?
>
> **Response:** We thank the reviewer for the suggestion. We plan to run more experiments on more GPUs with larger memory in the coming days, which will quantify how our attack performs with a larger number of clients.

---

> > ### Author Response · Authors · 2024-12-04
> > **Response to Reviewer Ykmd - Additional Experiments with Larger Number of Clients**
> >
> > We have conducted additional experiments based on the reviewer's comments to quantify the attack's performance with more than eight clients. We evaluated the attack with the maximum number of clients supported by our hardware during the rebuttal period, 32 for LeNet and 16 for ResNet9 and ResNet18. The attached figure shows that the LPIPS of the reconstructed images increases in proportion to the number of clients for all three models. We also observe a more gradual increase in LPIPS for ResNet9 and ResNet18 compared to LeNet, though the absolute values are higher. We use our default hyperparameters of 16 images per client, 3 local iterations, and an inversion learning rate of 0.1 to obtain these results. We also provide visualizations of how the image reconstruction quality is affected by larger numbers of clients for each model. These results indicate that the attack can meaningfully reconstruct images with 16-32 participants, which is near the upper end for cross-silo FL applications.
> >
> > Our results can be viewed at: https://anonymous.4open.science/r/CuriousClient_experiments-3701

---

### Author Response · Authors · 2024-11-14
**Code and Model Release**

Here is the repository link for "Federated Learning Nodes Can Reconstruct Peers' Image Data". It includes all the code and pretrained models necessary to replicate the curious client attack, postprocessing, and masked diffusion enhancer (MDE).

https://anonymous.4open.science/r/curiousclient-5B6F

---

### Meta-Review · Area_Chair_57JJ · 2024-12-16

**Metareview:**

The paper proposes and demonstrates a gradient inversion attack in federated learning by a client node against other client nodes.
Demonstrating an attack in a novel setting is potentially very interesting.
The reviewers criticise the paper for unrealistic assumptions and limited methodological novelty.
The authors' response seeks to address the reviewer concerns, but mostly fails to make substantial changes.
Unlike many of the reviewers, I agree with the authors' view that demonstrating an attack in the client-vs-client scenario is potentially interesting. However, I feel that the current manuscript provides a very incomplete view of the risk, as the attack requires highly unrealistic assumptions of a client being able to guess certain parameters of the algorithm. In order to demonstrate the authors' claim, it would be important to conduct additional experiments to evaluate the sensitivity of the method to these assumptions. As the current manuscript fails to provide this insight and mainly demonstrates the feasibility of the attack in an ideal scenario, I feel it is not ready for publication at ICLR.

**Additional Comments On Reviewer Discussion:**

There was limited participation from the reviewers in the discussion so I have evaluated the author responses and review myself.

---

### Decision · Program_Chairs · 2025-01-22

Reject